Manuscript prepared for Geosci. Model Dev.
with version 2015/04/24 7.83 Copernicus papers of the LaTeX class copernicus.cls.
Date: 17 November 2016

# Optimal numerical solvers for transient simulations of ice flow using the Ice Sheet System Model (ISSM versions 4.2.5 and 4.11)

Feras Habbal[1] , Eric Larour[2] , Mathieu Morlighem[3] , Helene Seroussi[2] ,
Christopher P. Borstad[4] , and Eric Rignot[2,3]

[1]University of Texas Institute for Geophysics, J.J. Pickle Research Campus, Building 196, 10100 Burnet Road (R2200), Austin, TX 78758-4445, USA
[2]Jet Propulsion Laboratory - California Institute of technology, 4800 Oak Grove Drive MS 300-323, Pasadena, CA 91109-8099, USA
[3]University of California, Irvine, Department of Earth System Science, Croul Hall, Irvine, CA 92697-3100, USA
[4]Department of Arctic Geophysics, University Centre in Svalbard, Longyearbyen, Norway

*Correspondence to:* Feras Habbal (ferashabbal@utexas.edu)

**Abstract.**

Identifying fast and robust numerical solvers is a critical issue that needs to be addressed in order to improve projections of polar ice sheets evolving in a changing climate. This work evaluates the impact of using advanced numerical solvers for transient ice-flow simulations conducted with the JPL/UCI Ice Sheet System Model (ISSM). We identify optimal numerical solvers by testing a broad suite of readily available solvers, ranging from direct sparse solvers to preconditioned iterative methods, on the commonly used Ice Sheet Model Intercomparison Project for Higher-Order ice sheet Models benchmark tests. Three types of analyses are considered: mass transport, horizontal stress balance, and incompressibility. The results of the fastest solvers for each analysis type are ranked based on their scalability across mesh size and basal boundary conditions. We find that the fastest iterative solvers are $\sim$1.5–100 times faster than the default direct solver used in ISSM, with speed-ups improving rapidly with increased mesh resolution. We provide a set of recommendations for users in search of efficient solvers to use for transient ice-flow simulations, enabling higher-resolution meshes and faster turnaround time. The end result will be improved transient simulations for short-term, highly resolved forward projections (10–100 year time scale) and also improved long-term paleo-reconstructions using higher-order representation of stresses in the ice. This analysis will also enable a new generation of comprehensive uncertainty quantification assessments of forward sea-level rise projections, which rely heavily on ensemble or sampling approaches that are inherently expensive.

## 1 Introduction

Fast and efficient numerical simulations of ice flow are critical to understanding the role and impact of polar ice sheets (Greenland Ice Sheet, GIS, and Antarctica Ice Sheet, AIS) on sea-level rise in a changing climate. As reported in the Intergovernmental Panel on Climate Change AR5 Synthesis report (Pachauri et al., 2014), "The ability to simulate ocean thermal expansion, glaciers and ice sheets, and thus sea level, has improved since the AR4, but significant challenges remain in representing the dynamics of the Greenland and Antarctic ice sheets." One of these challenges is the fact that Ice Sheet Models (ISMs) need to resolve ice flow at high spatial resolution (500 m to 1 km) in order to capture mass transport through outlet glaciers. This is especially the case for the GIS, which has a significant number of outlet glaciers in the 5–10 km width range (Rignot et al., 2011; Morlighem et al., 2014; Moon et al., 2015). This leads to transient ice-flow simulations with highly resolved meshes, which in turn reduces the time step prescribed by the Courant-Friedrichs-Lewy (CFL) condition that is necessary to maintain convergence and avoid developing numerical instabilities. This combination of high spatial and temporal resolution implies that ISMs are faced with challenges involving both scalability and speed.

The traditional approach to address this combined challenge is to solve a simplified set of equations for stress balance, relying on approximations to the stress tensor, which drastically reduce the number of degrees of freedom (dofs). These approximations have been extensively documented in the literature (Hindmarsh, 2004) and will not be described in detail here. However, we provide a brief summary of the characteristics of these models in order to relate the implications of our results in terms of solver efficiencies. The most comprehensive system of equations for modeling stress balance in ice flow is the full-Stokes model (Stokes, 1845), which captures each component of the stress tensor, and is hence the most complete physical description of stress equilibrium. It comprises four dofs (i.e. three velocity components and pressure) that are solved on a 3D mesh.

The Higher-Order formulation (HO, Blatter, 1995; Pattyn, 2003) uses the fewest assumptions to the stress tensor. This model neglects horizontal gradients of vertical velocities by assuming that these terms are negligible compared to vertical gradients of horizontal velocities. In addition, bridging effects are neglected. The resulting model comprises two dofs for horizontal velocities that are solved on a 3D mesh. Subsequently, the vertical velocity is recovered using the incompressibility equation. The next simplified formulation, the Shallow-Shelf or Shelfy-Stream Approximation (SSA, MacAyeal, 1989), arises from further assuming that vertical shear is negligible. This results in a set of two equations for the horizontal components of velocity that are collapsed onto a 2D mesh, where the vertical velocity is recovered through the incompressibility equation. This is one of the most efficient models used for fast-flowing ice streams and ice shelves, where motion is dominated by sliding (MacAyeal, 1989; Rommelaere, 1996; MacAyeal et al., 1998).

Finally, for the interior of the ice sheet, ISMs rely on the Shallow Ice Approximation (SIA, Hutter, 1983). In this model, horizontal gradients of vertical velocity are neglected compared to the vertical

gradients of horizontal velocities and only the components of vertical shear are included in the deviatoric stress (i.e. $\sigma'_{xz}$ and $\sigma'_{yz}$). This reduces the stress balance equations to a simple analytical formula relating the surface slope, ice thickness, and basal friction at the ice/bedrock interface. It is computationally very efficient and has been relied upon for long paleo-reconstructions of ice from the Last Glacial Maximum (LGM) to present day (Payne and Baldwin, 2000; Ritz et al., 1996; Huybrechts, 2004).

This list of model approximations is not exhaustive and does not include hybrid approaches such as the L1L2 formulation that mixes both SSA and SIA approximations. For readers that are interested in this topic, a comprehensive classification can be found in Hindmarsh (2004). Increasingly though, simple approximations such as the SIA have proven incapable of replicating observed velocity changes, such as the rapid acceleration of the West Antarctic Ice Sheet (Rignot, 2008) in the past two decades, or seasonal variations in surface velocities exhibited by GIS outlet glaciers (Moon et al., 2015). In addition, they are unable to capture ice-flow dynamics at resolutions compatible with most of the GIS outlet glaciers and fast ice streams of the AIS. In this context, the need for leveraging faster solvers within ISMs using accurate ice-flow formulations is critical for improving short-term projections of sea-level rise.

The Ice Sheet System Model (ISSM) framework relies on a massively parallelized thermo-mechanical finite element ice sheet model that was developed to simulate the evolution of Greenland and Antarctica in a changing climate (Larour et al., 2012). ISSM employs the full range of ice-flow approximations described above, and is therefore a good candidate for studying the efficiency of different solvers on ice-flow models. By default, ISSM relies on a direct numerical solver called the MUltifrontal Massively Parallel sparse direct Solver (MUMPS, Amestoy et al., 2001, 2006), to solve the system of algebraic equations resulting from the finite element discretization of the transient evolution of an ice sheet (i.e. solving the discrete mass transport, momentum balance, and thermal equations).

Using a direct parallel solver provides a robust and stable numerical scheme. However, this approach tends to be slow and memory intensive for large problems, where the number of dofs approaches 100,000 or more. As noted by Larour et al. (2012), the CPU time consumed by the default solver (i.e. MUMPS) accounts for 95% of the total solution time. In addition, there are significant problems with scalability associated with the direct solver approach Larour et al. (2012), which have not been explored to date, that preclude ISSM from efficiently running large-scale, high-resolution projections for the GIS and AIS. In addition to MUMPS, ISSM can also use numerical methods provided by the extensive suite of PETSc solvers, including iterative methods combined with preconditioning matrices that are well suited for ice-flow simulations. In addition to MUMPS, ISSM can also use numerical methods provided by the Portable Extensible Toolkit for Scientific Computations (PETSc, Balay et al., 1997), including iterative methods combined with preconditioning matrices that are well suited for ice-flow simulations. In order to reduce the impact of the numer-

ical solver as the bottleneck on solution time, this study evaluates the performance of using state-of-the-art numerical solvers for transient ice-flow simulations. Our approach is to characterize the impact of using a suite of readily available PETSc solvers to accelerate ISSM simulations involving higher-order ice-flow formulations. Our goal is to identify fast and scalable solvers that are stable across different basal sliding conditions. Here, we conduct a comprehensive assessment of numerical solvers using calibrated test cases from the well-known Ice Sheet Model Intercomparison Project for Higher-Order ice sheet Models (ISMIP-HOM) benchmark experiments (Pattyn et al., 2008). Using these well-studied benchmark tests allows us to evaluate the performance of numerical solvers for ice-flow simulations employing the HO formulation in a repeatable manner.

This work specifically focuses on this widely used formulation, as it currently represents the most computationally demanding model (short of full-Stokes) capable of capturing vertical as well as horizontal shear stresses necessary to model an entire basin (Pattyn, 1996). The finite element discretization of the full-Stokes model leads to a well-studied saddle point problem, which represents an active area research in geophysics (e.g. Benzi et al. (2005); Elman et al. (2014)). While recent work (e.g. Isaac et al. (2015)) has shown promising results, stable iterative full-Stokes solvers are not readily available and, in general, are significantly disruptive to integrate in terms of their code base, which is the reason we will not be considering them in this study.

The HO model represents the next, most complete formulation and represents a significant computational bottleneck compared to its 2D and 1D counterparts, which are significantly less demanding because of the drastic reduction in the number of dofs required for vertically collapsed 2D meshes (SSA) or local 1D analytical formulations (SIA). In contrast to the studies focused on specific, customized solvers for approximate flow models (i.e. Brown et al. (2013); Cornford et al. (2013); Tezaur et al. (2015); Tuminaro et al. (2016)), this work surveys a broad range of solvers for the HO ice-flow model. While our analysis uses ISSM, our results are relevant to other ice-flow models and frameworks that use PETSc solvers.

The manuscript is structured as follows. In section 2, we describe the ISMIP-HOM experiments that we consider and the approach adopted for testing different numerical methods. In section 3, we summarize efficient baseline solvers for transient simulations using the ISSM framework. In section 4, we discuss the timing results from testing a wide range of solvers, which in addition to enabling large-scale simulations yields significant speed-ups in solution time. We then conclude on the scope of this study and summarize our findings.

## 2 Model and Setup

In order to identify optimal numerical solvers for a broad class of transient ice-flow simulations, we test a suite of PETSc solvers on synthetic ice-flow experiments with varying basal sliding conditions. We consider the effectiveness of competing solvers (in terms of speed) using the ISMIP-HOM

tests, since these experiments represent a suite of accepted benchmark tests that are commonly used in the community to validate higher-order (3D) approximations of the stress balance equations. We first use Experiment F of the ISMIP-HOM tests to evaluate competing numerical solvers since it entails a transient ice-flow simulation with two test cases involving distinct basal sliding regimes. This transient simulation allows us to independently test solvers on each analysis component (mass transport, horizontal stress balance, and incompressibility) underlying a transient simulation in ISSM and evaluate the performance of competing solvers for models using different basal sliding conditions. Experiment F is representative of the type of physics solved for in many scenarios of ice sheets retreating and advancing onto downward or upward-sloping bedrocks. It is therefore wide-ranging in terms of applicability and happens to be a commonly accepted benchmark experiment that is used by many ISMs. However, since Experiment F specifies a constant viscosity for ice, we also consider ISMIP-HOM Experiment A as it includes a nonlinear rheology for ice. While this is only a static test, Experiment A allows us to evaluate the performance of solvers applied to the horizontal stress balance equations for simulations using a more physically realistic model of ice rheology, albeit only for one basal boundary condition. For testing the impact of different basal sliding conditions on solver performance, we refer to the results from Experiment F, which includes both sliding and no-slip basal conditions.

Experiment F consists of simulating the flow of a 3D slab of ice (10 km square, 1 km thick) over an inclined bed (3 degrees) with a superposed Gaussian-shaped bump (100 m in height) until the free surface geometry and velocities reach steady state. Here, we run our transient simulation for 1500 years, using 3-year time steps, in order to allow the free surface to relax and reach a steady-state configuration. The prescribed material law is a linear viscous rheology that results in a constant effective viscosity for ice. In order to test different friction parameterizations, Experiment F explores two test cases of boundary conditions at the bedrock/ice interface: 1) no-slip (frozen bed) and 2) viscous slip (sliding bed). For both scenarios, we apply Dirichlet boundary conditions for the velocities along the boundary and set the values to zero. This is slightly different from using periodic boundary conditions suggested by the ISMIP-HOM benchmark tests; however, Dirichlet boundary conditions are more relevant to boundary conditions generally used by modelers. Fig. 1 displays the surface velocity and surface elevation results at the end of the transient simulation using ISSM. These results are consistent with typical steady state profiles for Experiment F, with slight differences near the boundaries affected by using different boundary conditions.

Experiment A simulates the flow of a 3D slab of ice (80 km square, 1 km thick) over an inclined bed (0.5 degrees) with sinusoidal bumps (500 m amplitude). This experiment assumes that the ice is frozen to the bed (i.e. no-slip boundary condition). While this test is prognostic in nature and does not consider the time-evolution of the ice configuration, it does include a nonlinear viscosity model for ice, which is more realistic than the constant viscosity specified in Experiment F. Similarly to

Experiment F, we prescribe Dirichlet boundary conditions for the velocities along the boundary and set the value to zero.

Our approach for identifying efficient numerical methods for each analysis component of the transient simulation in ISSM is to independently test combinations of preconditioning matrices with iterative methods on the system of equations resulting from the finite element discretization of the

stress balance and mass transport equations. Since we rely on the HO formulation, the stress balance only solves the horizontal stress balance and requires an additional step to solve for the vertical velocities. Here, we use the incompressibility equation and an $L_2$ projection to solve for the vertical velocities. We call these steps the horizontal velocity solution and incompressibility solution, respectively. In addition, running a transient simulation implies a mass transport module, which combined

with the velocity analyses requires three systems of equations to be solved at each time step.

For each system of equations, we test a wide range of solvers including the default solver (MUMPS) and preconditioned iterative methods provided by PETSc. When referring to the solvers available through the PETSc interface, we rely on the abbreviations used in the PETSc libraries by labeling a preconditioning matrix as a PC and an iterative method as a KSP (Krylov subspace method).

Here a preconditioning matrix improves the spectral properties of the problem (i.e. the condition number) without altering the solution provided by the iterative method. Since the Jacobian of the system of equations resulting from the finite element discretization of the horizontal stress balance is symmetric positive definite a wide range of iterative solvers and preconditioners are applicable and potentially efficient. For a complete review of potential solvers we point to Benzi et al. (2005)

and Saad (2003). In the subsequent benchmark simulations, 10 PC matrices, and 20 KSP iterative methods are tested in unique solver combinations. Additionally, the effect of not applying a preconditioning matrix to the iterative method is tested for each KSP represented by PC=none. The solvers tested for all analysis types are indicated by the permutations of the KSP and PC methods listed in the headings of 4. In an attempt to use the PETSc solvers in ISSM with minimal invasiveness, we

restrict the inclusion of KSPs and PCs from the PETSc suite by only testing methods that naturally fit the ISSM framework (i.e. without the need for customization or specialization of the solver routine). Anticipating that modelers may not tune the individual components in PETSc, we test each method using default values to evaluate baseline performance provided by each method natively.

The slab of ice in Experiment F is modeled using four levels of mesh refinement. The smallest,

coarsest-resolution model consists of 2000 elements resulting from a $10 \times 10 \times 10$ $(x, y, z)$ grid of triangular prismatic 3D elements. Three larger models are produced by refining each direction of the smallest model by a factor of 2, leading to 16,000, 128,000, and 1,024,000 element models. Each model size is tested using four CPU cases: 250, 500, 1000, and 2000 elements per CPU. Only the fastest timing results for simulations where the solution passes three ISSM convergence tests (i.e.

mechanical stress balance and convergence of the solution in both a relative and absolute sense) at each time step using default tolerances are included in the ranking results. All of these simulations

were performed on the NASA Advanced Supercomputing Pleiades cluster (Westmere nodes: 2 six-core Intel Xeon X5670 processors per node, 24 GB per node) using ISSM version 4.2.5 and PETSc version 3.3.

To study the impact of using a nonlinear viscosity model for ice on solver speed and convergence, we follow the same methodology applied to Experiment F (i.e. same solvers, discretization strategy, and CPU cases) and evaluate the performance of solvers applied to the stress balance equations for Experiment A. However, we omit testing the largest model size (i.e. 1,024,000 elements) due to the intense computational resources necessary for this model size and the limited information gained by this prognostic test relative to the more comprehensive transient model. Simulations of Experiment A were performed more recently on the Pleiades cluster using upgraded Broadwell nodes (2 fourteen-core Intel Xeon E5-2680v4 processors per node with 128 GB per node) with ISSM version 4.11 and PETSc version 3.7. Updates to the ISSM code from version 4.2.5 to version 4.11 have added new capabilities that are not used in this study. The solution methods and algorithms between these versions are the same, and the results from this study apply to all intermediate versions that users may be working with.

## 3  Results

Since our primary interest is identifying fast, stable solvers for transient ice-flow simulations, we first present the results from Experiment F. Using the profiling features in ISSM, we evaluate the timing results for each simulation (measured in seconds), which consists of the CPU time associated with assembling the stiffness matrix, load vector, solving the system of equations, and updating the input from the solution. Since this study focuses on the relative performance of the tested solvers, the timing results for different methods are directly comparable, as the additional steps are consistent across the tested methods and do not bias the results. Our timing results, measured in seconds, consists of the CPU time associated with assembling the stiffness matrix, load vector, solving the system of equations, and updating the input from the solution. Only the fastest results for each model size for solving the horizontal velocity analysis (fastest 10%), the incompressibility analysis (fastest 5%), and the mass transport analysis (fastest 5%) are shown in Figs. 2–4, respectively. These thresholds (i.e. 10%, 5%, and 5%) are chosen so as to exhibit clear trends in identifying the fastest and most robust solvers. Here, we associate the robustness of a solver (PC/KSP combination) in terms of efficiently solving a given analysis across the wide range of model sizes and both basal boundary conditions. This classification is different from solvers that are optimal (i.e. fastest) for a specific case, but it allows modelers to identify solvers that are fast across the largest set of conditions, be it mesh size, number of available CPUs, or basal sliding regime. Modelers interested in optimal performance for a specific simulation should consult Figs. 2–4 for each analysis component and use

a solver corresponding to a color-filled symbol (i.e. fastest 1% result) closest to their model size, where the number of recommended CPUs is specified by the color of the symbol.

For Experiment F, we highlight the most robust solvers in Figs. 2–4 using red boxes. Thus, a red box highlights a solver (PC/KSP combination) where all four symbols (i.e. all tested mesh sizes) are among the fastest methods for both basal boundary conditions. Whereas color-filled symbols identify the solvers that are among the fastest timing results (top 1%) for that mesh size and basal boundary condition. The highlighted solvers from Figs. 2–4 may be used as ISSM solver defaults for each analysis type of the transient simulation (i.e. horizontal velocity, incompressibility, and mass transport). For the horizontal velocity solution, the results in Fig. 2 show six highlighted solvers are robust (i.e. four symbols displayed for both basal boundary conditions). These results indicate that using a block Jacobi preconditioner is well suited for this analysis type for both sliding and frozen bed scenarios. For the incompressibility analysis, the highlighted solvers in Fig. 3 indicate that using a variant of the Jacobi preconditioner (block Jacobi, Jacobi or point block Jacobi), in conjunction with the corresponding KSPs yields the most robust results. For the mass transport analysis, the situation is more nuanced in terms of preconditioners, but both the bcgs and bcgsl KSP solvers tend to be robust across several preconditioners. Surprisingly, not using a preconditioner for the mass transport analysis seems to yield very fast and robust results when used in combination with the lsqr and bcgs solvers, which was not expected.

Figs. 5 and 6 plot the weak and strong scalability associated with solving Experiment F using the default ISSM solver (MUMPS) and iterative solvers selected from the highlighted solvers in Figs. 2–4 for each analysis type underlying the transient simulation. Specifically, we compare the default solver results to a combined strategy that uses a point block Jacobi preconditioner with a biconjugate gradient stabilized iterative method to solve the mass transport analysis (PC=pbjacobi, KSP=bcgsl), a block Jacobi preconditioner with a minimum residual iterative method to solve the horizontal velocity analysis (PC=bjacobi, KSP=minres), and a point block Jacobi preconditioner with a conjugate gradient iterative method applied to the normal equations to solve the incompressibility analysis (PC=pbjacobi, KSP=cgne). One issue that arose while carrying out the weak scalability analysis was that simulations using MUMPS to solve the largest model (i.e. 1,024,000 elements) experienced memory and cluster issues for both sliding and frozen bed scenarios (e.g. computational nodes restarting and general memory issues). For these tests, we estimate the total time required to solve the largest model with the MUMPS solver by linearly extrapolating the total time from the number of iterations completed during a two-hour and eight-hour run. These estimated timing results are displayed by diamond symbols in Fig. 5 for the direct solver only.

In considering the magnitude of the slopes representing the weak and strong scalability, we recall that our timing results include routines outside of the solver procedure (i.e. assembling the stiffness matrix, load vector, and updating the input from the solution) that are not necessarily scalable. However, the relative scalability (i.e. differences in slope) between the preconditioned iterative methods

and the direct solver illustrates the differences in performance between these approaches. Optimal weak scalability for a solver implies a horizontal slope in Fig. 5 and the ability to solve increasingly refined models with a fixed ratio of elements per CPU in constant time. Here, the slopes representing the weak scalability of the preconditioned iterative solver for the frozen bed and sliding bed configurations are 0.441 and 0.495, respectively. Whereas the slopes for the direct solver are much larger at 1.124 and 1.165 for the frozen and sliding bed configurations, respectively. For the largest model (i.e. 1,024,000 elements) the iterative solver is more than two orders of magnitude faster than the ISSM default solver: ∼57 hours (estimated) compared to ∼15 minutes. As 5 indicates, using a preconditioned iterative method over the direct solver is increasingly beneficial for larger model sizes. For very small models (i.e. 2000 elements), using MUMPS is marginally slower (∼1.5 times) than the presented iterative methods. Optimal strong scalability implies a slope equal to -1 in Fig. 6 and the ability to solve a model with a fixed number of elements faster by using more CPUs. The slopes in Fig. 6 representing the strong scalability of the direct solver for the frozen and sliding bed configurations are -0.332 and -0.399, respectively. In comparison, the slopes for the combined iterative solvers applied to the frozen and sliding bed configurations are -0.897 and -0.911, respectively, clearly favoring these solvers over the direct solver.

To show the impact of nonlinear viscosity on the efficiency of the presented solvers, we plot the timing results for solving the stress balance equations in Experiment A (Fig. 7). Fig. 7 plots the fastest (top 15%) timing results for each mesh size, using the same symbols as the previous plots for Experiment F, where color-filled symbols represent the overall fastest results (i.e. top 1%) for each model size. In comparing our results to using the default ISSM solver (MUMPS), we plot the strong and weak scalability (Fig. 8) for the direct solver and one of the fastest solvers identified from Fig. 7 (KSP=cg, PC=bjacobi). Similar to the results for Experiment F, the slopes of the weak scalability (Fig. 8a) for the preconditioned iterative method (0.205) is also much smaller (i.e. closer to optimal scalability) than the direct solver (0.883). In comparing the strong scalability of these solvers (Fig. 8b), the slope of the preconditioned iterative method (-0.737) also indicates better performance than the slope of the direct solver (-0.270).

## 4 Discussion

Solving the horizontal velocity analysis dominates the CPU time needed to solve a transient simulation since this analysis involves more dofs and has a much higher condition number than the mass transport and incompressibility analyses. Our results, however, show that this bottleneck can be significantly reduced for moderate-sized models (i.e. 16,000 to 128,000 elements) by using any of the highlighted solvers, which leads to significant speed-ups relative to the default solver (i.e. ∼7.5-37 times faster). As Fig. 5 shows, using a direct solver such as MUMPS is not recommended for transient simulations of models using more than 128,000 elements. This is both due to the sig-

nificant speed-ups (more than 10 times) achieved by using iterative solvers for transient simulations involving large models (more than 20,000 elements) and to the inherent memory restrictions associated with using the direct solver that prevent massive transient simulations (more than 1,000,000 elements).

Most of the limitations associated with using the default solver on large models arise from the LU Factorization phase in the MUMPS solver, which is not yet parallelized. This could be remedied by switching on the out-of-core computation capability for this decomposition, but this has not been successfully tested yet and would potentially shift the problem of memory limitations to disk space and read/write speeds (the size of the matrices being significant). Furthermore, Fig. 5 indicates that the highlighted solvers are not only capable of handling the largest model (1,024,000 elements), but the solution time is nearly equivalent to using the default MUMPS solver on a significantly smaller model size (i.e. ∼20,000 elements).

In evaluating the effect of using a nonlinear viscosity model for ice on solver performance, we see that many of the methods which efficiently solve the horizontal velocity analysis for Experiment A (Fig. 7) are consistent with the solvers highlighted for Experiment F (Fig. 2), which includes a much simpler constant viscosity for ice. Specifically, we see that the block Jacobi preconditioner (PC=bjacobi) is effective across a number of iterative methods for both benchmark experiments. While this comparison only extends up to model sizes of 128,000 elements, we see from the plot of weak scalability (Fig. 8a) that using the iterative solver results in speed-ups ranging from ∼1.2–19 times faster than using the default solver for model sizes increasing from 2,000–128,000 elements.

In practice, users may experience issues with numerical convergence when applying some of the iterative methods presented in Figs. 2–4 for their particular application. In these instances using the ISSM default solver (MUMPS) provides a stable solution strategy. Furthermore, since solving the horizontal velocity analysis is the most CPU-time intensive stage of the transient simulation process, using a direct solver for the other analysis types and relying on Fig. 2 to select an optimal solver for the horizontal velocity analysis may provide the best balance between stability and speed.

While the relative rankings of the tested solvers presented in this work are specific to the ISMIP-HOM Experiments, applying these methods to simulations using realistic model parameterizations (e.g. data-driven boundary conditions, anisotropic meshes, and complex geometries) also results in significant speed-ups compared to the default solver, though these computations are not shown here. We acknowledge that in relation to using synthetic test cases, real-world model parameterizations may affect the convergence and relative performance of the iterative solvers tested in this work. However, since any of the highlighted solvers are significantly more efficient than using a direct solver, our results provide a useful starting point for modelers looking for efficient methods to use for specific ice-flow simulations.

We recommend that future refinement of these results include customization of the PETSc components, which can lead to significant performance gains over the default values, and include more

realistic geometries with varying degrees of anisotropy. Finally, it should be noted that the presented optimal solvers do not require a supercomputer and may be used with fewer CPUs than the number indicated by the symbol color in Figs. 2–4. Indeed, the highlighted iterative methods may provide speed-ups (compared to using MUMPS) larger than Fig. 5 indicates when using computers with limited memory.

## 5  Conclusions


The results presented herein offer guidance for selecting fast and robust numerical solvers for transient ice-flow simulations across a broad range of model sizes and basal boundary conditions. Here, the highlighted solvers offer significant speed-ups ($\sim$1.5–100 times faster) relative to the default solver (MUMPS). Furthermore, the highlighted solvers enable large-scale, high-resolution transient

simulations that were previously too large to run with the default solver in ISSM. These combined benefits are consistent with results across a broad range of computational disciplines, which also show that iterative solvers are significantly more efficient than direct solvers for solving sparse linear systems as the number of dofs becomes large. While modelers may prefer to use a direct solver as a stable strategy, the significant performance gains attained using the preconditioned iterative meth-

ods highlighted in this study provide a compelling case to consider. Here, taking the time to find an efficient solver is strongly recommended for computationally demanding simulations involving high-resolution meshes as well as uncertainty quantification studies or parameter studies entailing repeated simulations.

## 6  Code Availability

The results from this work are reproducible using ISSM (versions 4.2.5–4.11) with the corresponding PETSc solvers used for each analysis type. Here, the current version of ISSM is available for download at https://issm.jpl.nasa.gov, and previous versions are available from the svn repository. The models for simulating these ISMIP-HOM Experiments are documented on the website and included in the test directory of the download.

*Acknowledgements.* This work was performed at the Jet Propulsion Laboratory, California Institute of Technology, and at the Department of Earth System Science, University of California Irvine, under two contracts, one a grant from the National Science Foundation (NSF) (Award ANT-1155885) and the other a contract with the National Aeronautics and Space Administration (NASA), funded by the Cryospheric Sciences Program and the Modeling Analysis and Prediction Program. Resources supporting the numerical simulations were provided

by the NASA High-End Computing (HEC) Program through the NASA Advanced Supercomputing (NAS) Division at Ames Research Center. We would like to acknowledge the insights and help from Dr. Jed Brown, as well as helpful feedback from the anonymous reviewers.

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

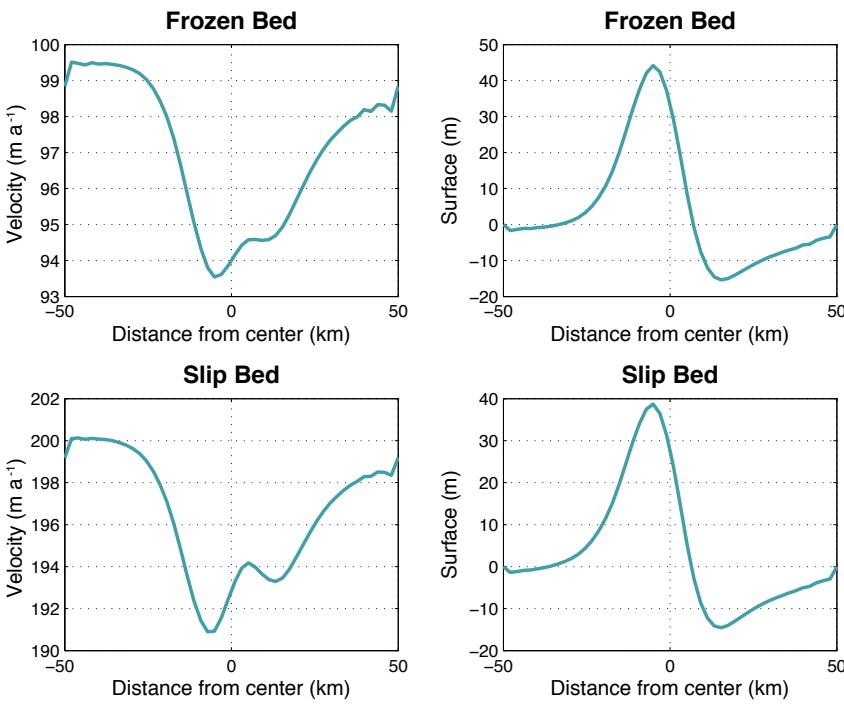

**Figure 1.** ISSM results for the ISMIP-HOM benchmark Experiment F transient simulation after 1500 years. Surface velocity (m a$^{-1}$) and steady state surface elevation profile (m) along the central flowline are shown for the frozen and sliding bed cases.

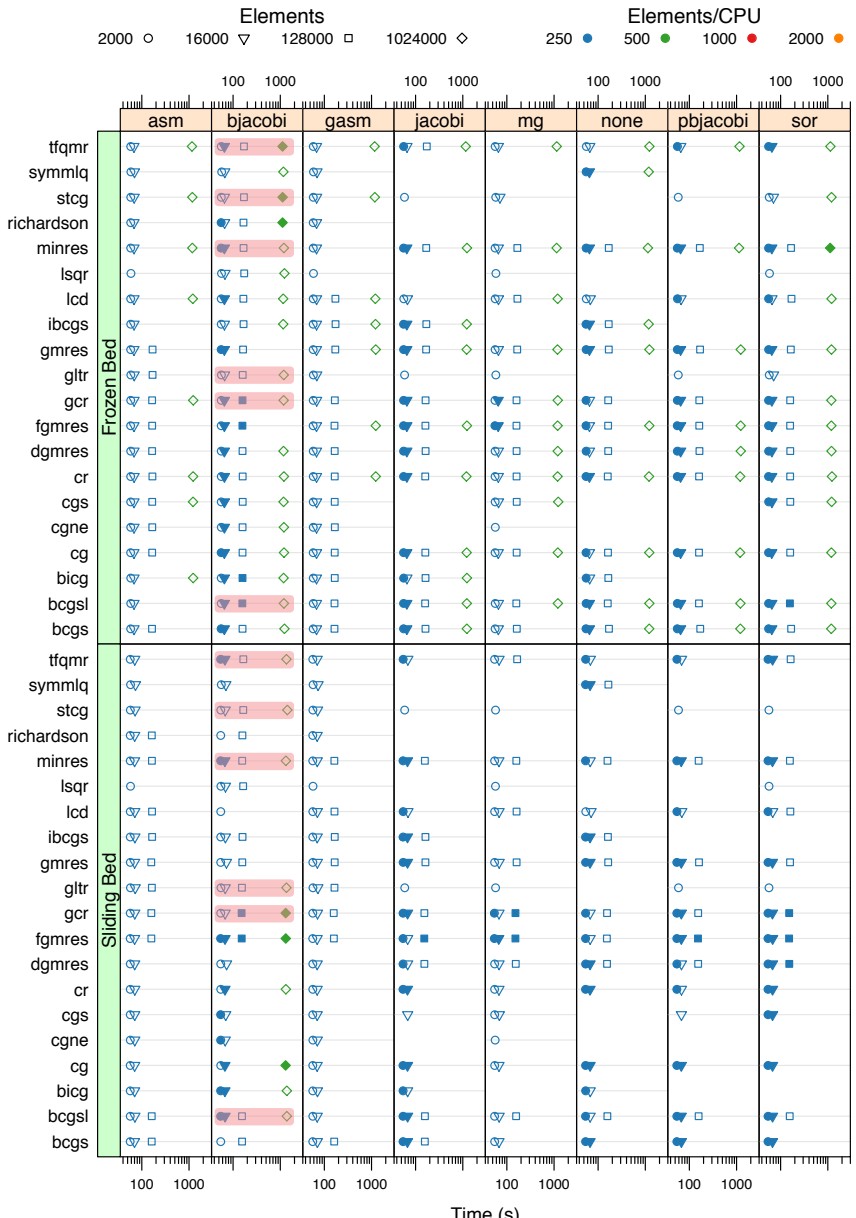

**Figure 2.** Horizontal velocity analysis: timing results for the fastest solvers (top 10%) tested on ISMIP-HOM Experiment F. The top (1%) timing results are distinguished using color-filled symbols. Both basal boundary conditions for Experiment F are shown: frozen bed (upper half) and sliding bed (lower half). Each solver is represented by the combination of a preconditioner (horizontal rows) and a Krylov subspace method (vertical columns) using PETSc abbreviations. Simulations are performed using four mesh sizes (denoted by the symbols in the legend) and four CPU cases (denoted by the colors in the legend). Only the fastest CPU case (i.e. color) is displayed. Red boxes highlight solver combinations that rank among the fastest methods for all model sizes and both bed conditions (i.e. four symbols in the top and bottom frame).

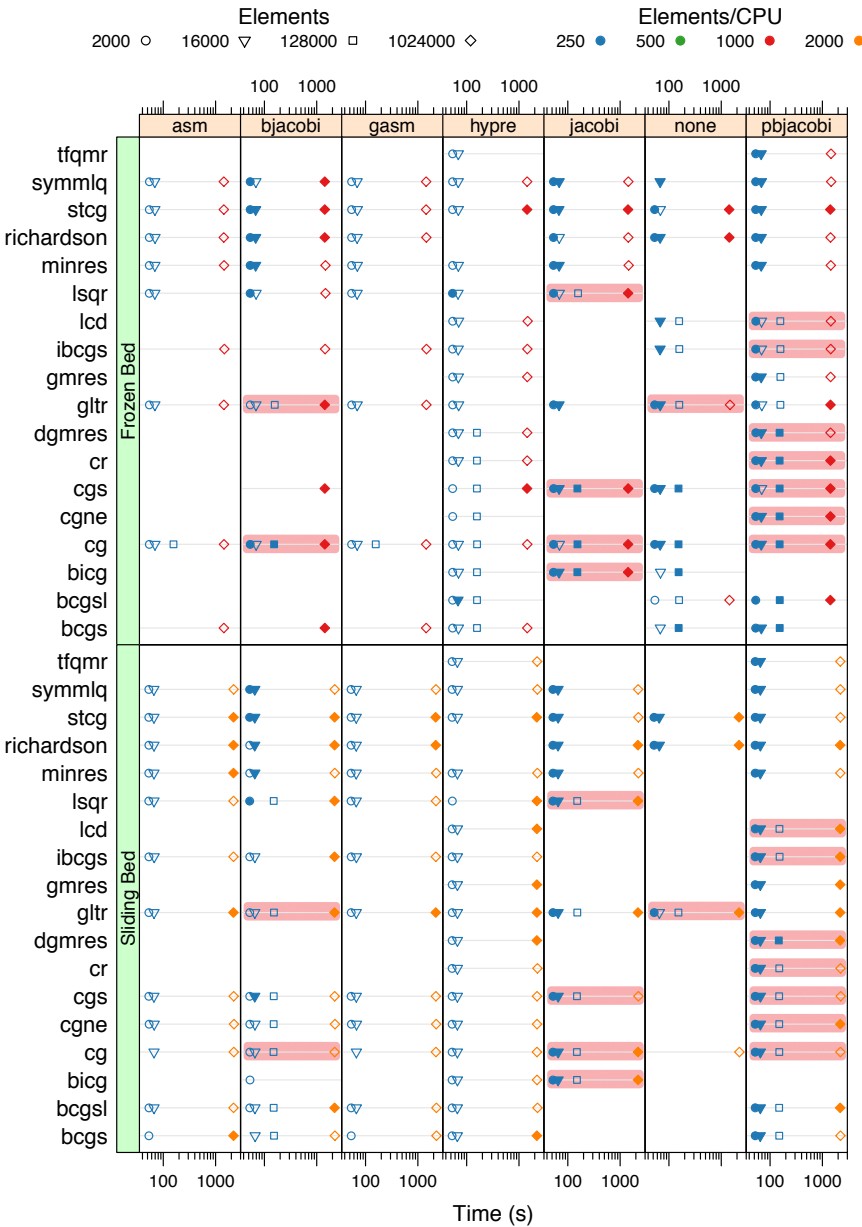

**Figure 3.** Incompressibility analysis: timing results for the fastest solvers (top 5%) tested on ISMIP-HOM Experiment F. The top (1%) timing results are distinguished using color-filled symbols. Red boxes highlight solver combinations that rank among the fastest methods for all model sizes and both bed conditions (i.e. four symbols in the top and bottom frame). See Fig. 2 for more details.

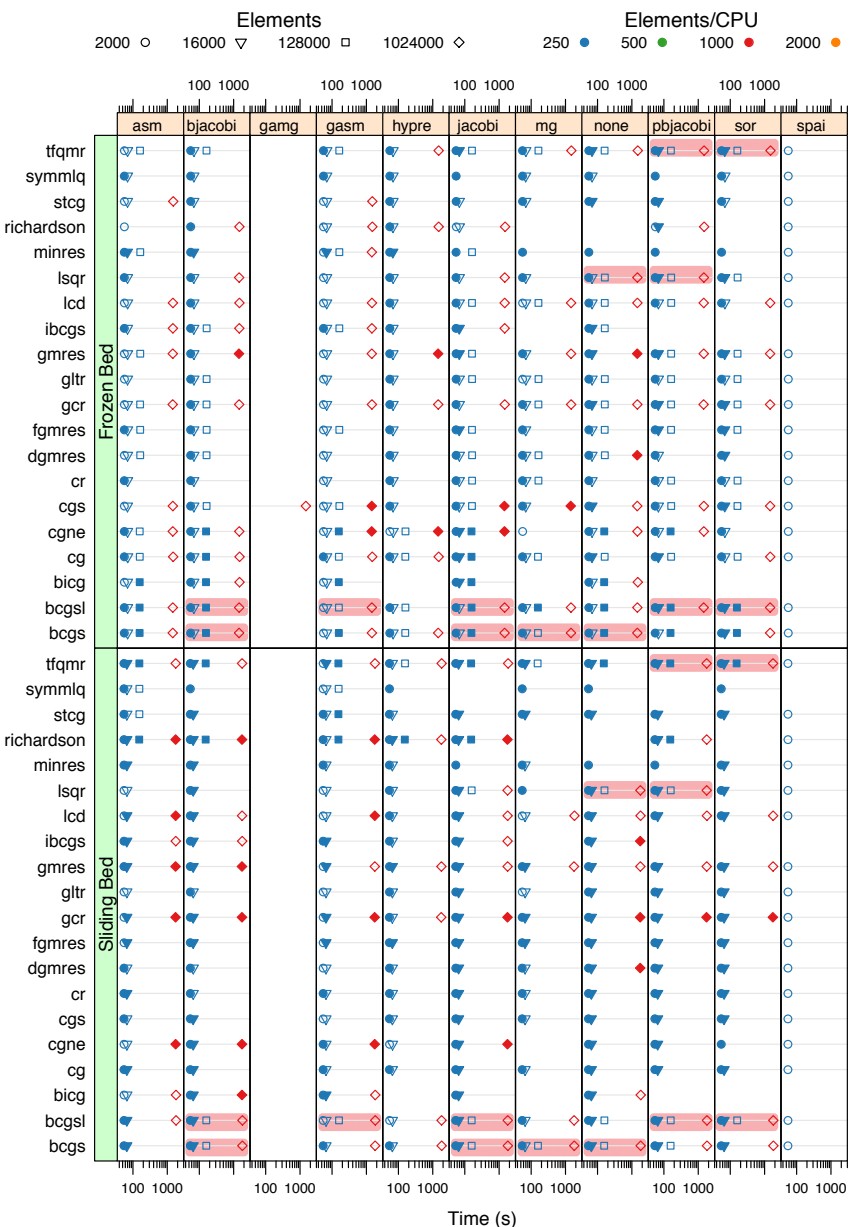

**Figure 4.** Mass transport analysis: timing results for the fastest solvers (top 5%) tested on ISMIP-HOM Experiment F. The top (1%) timing results are distinguished using color-filled symbols. Red boxes highlight solver combinations that rank among the fastest methods for all model sizes and both bed conditions (i.e. four symbols in the top and bottom frame). See Fig. 2 for more details.

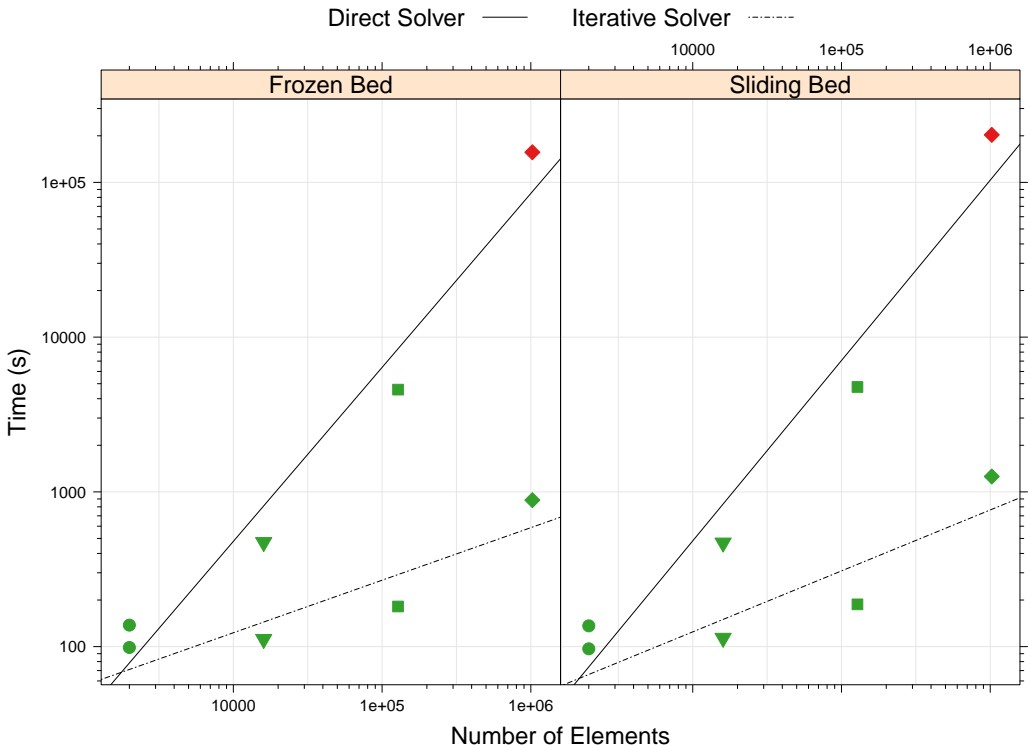

**Figure 5.** Weak scalability for simulating ISMIP-HOM Experiment F using the default ISSM solver (MUMPS) compared with a combination of robust solvers (selected from the highlighted solvers in Figs. 2–4) for each analysis component of the transient simulation. This combination consists of a point block Jacobi (pbjacobi) preconditioned biconjugate gradient stabilized (bcgsl) iterative method for the mass transport analysis, a block Jacobi (bjacobi) preconditioned minimum residual (minres) iterative method for the horizontal velocity analysis, and a point block Jacobi (pbjacobi) preconditioned conjugate gradient on the normal equations (cgne) for the incompressibility analysis. These simulations are conducted using a constant ratio of 500 elements per CPU (except for the largest model with the direct solver) and show the impact of increasing mesh size on simulation time (seconds). Ideal weak scaling is consistent with a horizontal slope. Timing results include the CPU time associated with assembling the stiffness matrix, load vector, solving the system of equations, and updating the input from the solution.

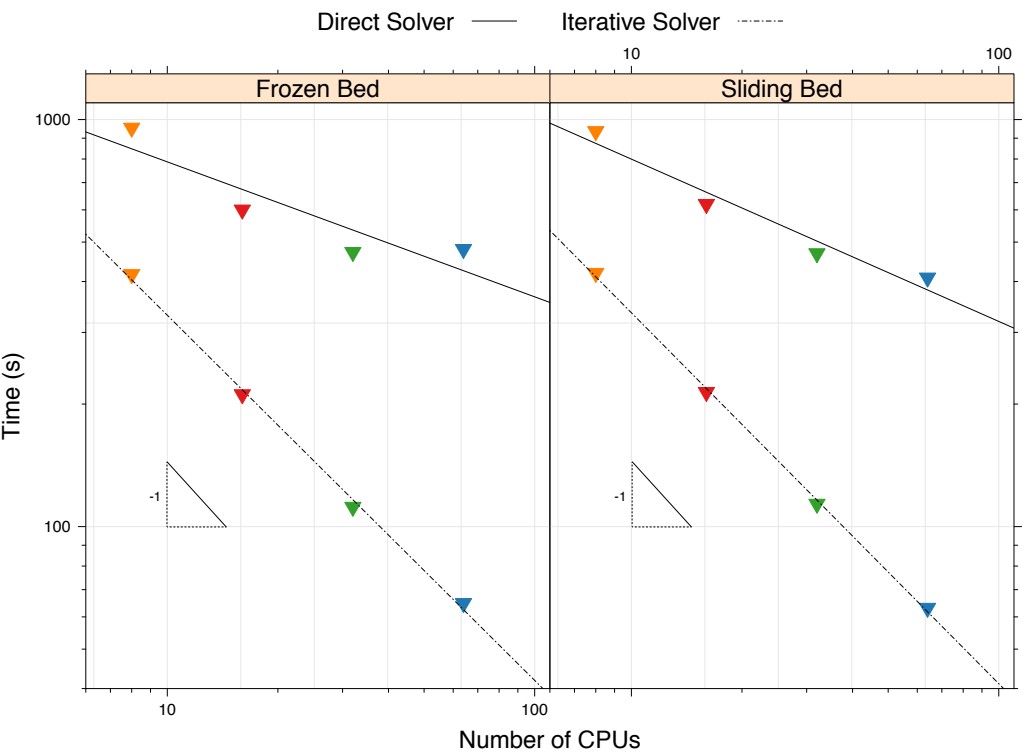

**Figure 6.** Strong scalability of the default ISSM solver (MUMPS) compared with a combination of robust solvers (selected from the highlighted solvers in Figs. 2–4) for the components of the transient ISSM simulation of ISMIP-HOM Experiment F. See Fig. 5 for the solvers specified for each analysis component. Strong scalability represents the impact of increasing the number of CPUs while keeping the mesh size constant (16,000 elements). Ideal strong scalability is consistent with a slope equal to -1. Timing results include the CPU time associated with assembling the stiffness matrix, load vector, solving the system of equations, and updating the input from the solution.

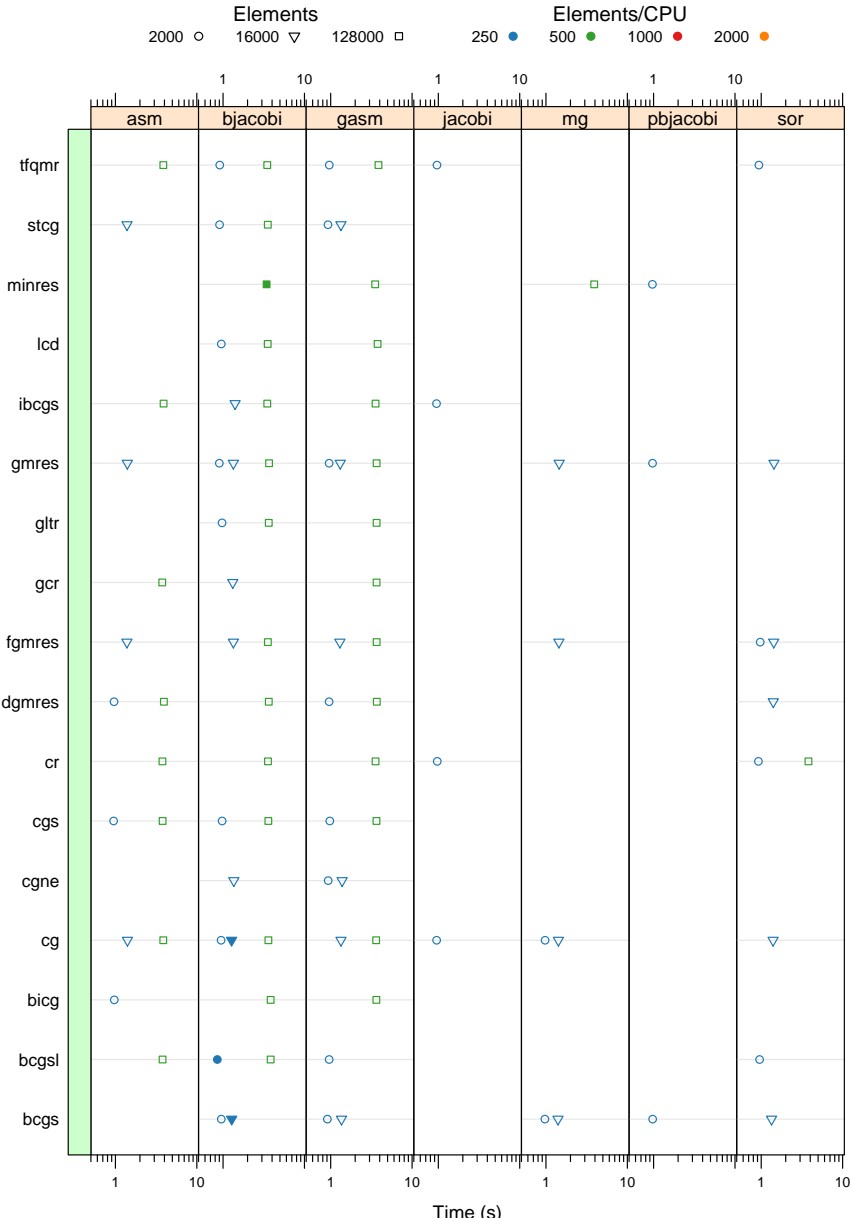

**Figure 7.** Horizontal velocity analysis: timing results for the fastest solvers (top 15%) tested on ISMIP-HOM Experiment A. The top (1%) timing results are distinguished using color-filled symbols. See Fig. 2 for more details.

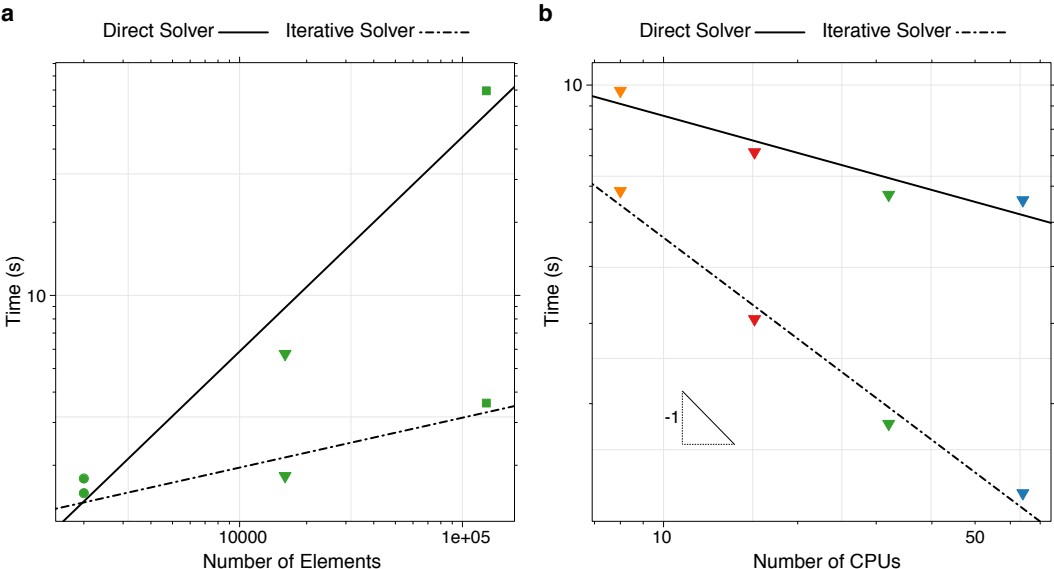

**Figure 8.** Scalability of the default ISSM solver (MUMPS) compared with a preconditioned iterative method (PC=bjacobi, KSP=cg) for ISMIP-HOM Experiment A. **a**, Weak scalability of solvers using a constant ratio of 500 elements per CPU; ideal weak scalability is represented by a horizontal slope. **b**, Strong scalability of solvers for a 16,000 element model; ideal strong scalability is represented by a slope equal to -1. Timing results include the CPU time associated with assembling the stiffness matrix, load vector, solving the system of equations, and updating the input from the solution.