# Peer review of "Optimal numerical solvers for transient simulations of ice flow using the Ice Sheet System Model (ISSM versions 4.2.5 and 4.11)"

_Geoscientific Model Development, 2016_

## Short Comment (SC1) · 6 Jun 2016

Dear authors,

In my role as Executive editor of GMD, I would like to bring to your attention our Editorial version 1.1:

http://www.geosci-model-dev.net/8/3487/2015/gmd-8-3487-2015.html

This highlights some requirements of papers published in GMD, which is also available on the GMD website in the 'Manuscript Types' section:

http://www.geoscientific-model-development.net/submission/manuscript_types.html

In particular, please note that for your paper, the following requirement has not been met in the Discussions paper:

- "The main paper must give the model name and version number (or other unique identifier) in the title."

Please provide a version number for ISSM in the title upon your revised submission to GMD.

Yours,

Astrid Kerkweg

---

## Referee Comment (RC1) · Anonymous Referee #1 · 11 Jul 2016

In this work, the authors compare the performances of several different direct and iterative solvers, provided by MUMPS and PETSc libraries respectively, for solving the transient ice flow model using ISSM. Specifically, the authors target a well known transient benchmark problem (ISMIP-HOM, test F), in the case of a frozen bed or sliding bed. The flow model is constituted by the ice velocity part (Blatter-Pattyn model with constant viscosity), a part for reconstructing vertical velocities, and the mass-transport part. The authors highlight some of the solvers that perform better on different mesh resolutions, for both frozen or sliding bed.

The detailed comparison of the solvers available in ISSM is certainly useful for the several ISSM users.

[Figure]

However, I have a few reservations about the impact that this work can have on a broader community.

- The benchmark problem addressed in this work has several simplifications that makes it not very representative of real problems, most notably: 1. Geometry is very simple (in constrast with complex margins or bed roughness encountered in real ice sheets). 2. Viscosity is constant, making the model linear. In real problems viscosity strongly depends on velocity and temperature, which makes the problem much harder to be solved numerically. 3. A relatively high basal friction coefficient is considered, which is not representative of what can be found in ice streams and ice shelves.

- The authors consider only off-the-shelf solvers that "naturally fit the ISSM framework", whereas several efforts (not mentioned by the authors) have been done in recent years in order to build efficient solvers/preconditioners tailored on the ice sheet problems. Some of these solvers have been demonstrated on large-scale simulations of Greenland or Antarctic ice sheets. See, for example, T. Isaac et al., SIAM J. Sci. Comput., 37(6), B804–B833; Tezaur et al., Procedia Computer Science, 51:2026-2035, ICCS, 2015, S. Cornford at al. J. Comput. Phys, 232(1):529-549, 2013; plus the one by Brown et al. already cited, but not discussed, by the authors.

I recommend that the authors make it clear in the abstract that they are only considering the off-the-shelf solvers readily available in ISSM. I also recommend to consider more realistic problems and to mention relevant work in the literature.

Minor comments:

- At line 144 the authors mention that they apply single-point constraints on velocity and thickness equations. I'd like the author to expand on this, mentioning how/with what values they constrain in a single point the velocity and thickness. Typically, single-point constraining is used in presence of a singular problems (which should not happen here), and it is known to artificially modify the spectrum of the matrix, which in turn can deteriorate the convergence of iterative solvers.

[Figure]

- Line 150, how the vertical velocity is reconstructed? With an L2 projection?

- The time reported in the tables is solver time, or total time (including assembly, linear solvers and I/O)? The weak scaling results for the iterative solvers are not very good and it would be useful to understand what is causing this.

―――――――――――――――――

---

## Referee Comment (RC2) · Anonymous Referee #2 · 25 Jul 2016

This work aims to provide insight into different solver choices for a particular full-Stokes ice sheet model (ISSM) by testing a range of iterative solver choices from the widely-available PETSc solver library on a specific test problem (ISMIP-HOM, experiment F) and contrasting with their native/default direct-solve approach (which uses MUMPS). They conclude that switching to the PETSc iterative solvers generally improves time-to-solution and scaling as the problem size (number of elements in the finite-element mesh) increases, and are able to provide some suggestions as to which solvers appear to be better suited to their needs. In my opinion, this is a useful contribution to the literature, and I found it to be well-written and well-organized. I do have a few suggestions which I think would greatly increase the usefulness of this work.

[Figure]

My primary criticisms, if you can call them that, are regarding the choice of benchmark problem. While I think that the choice of ISMIP-HOM problem F is a reasonable choice for representing a fully- or mostly-grounded ice sheet (like the Greenland Ice Sheet), I wonder how extendable the results and conclusions are to systems with fast-flowing ice streams and large dynamic ice shelves as are found in Antarctica, represented, for example, by the MISMIP family of benchmarks. In our experience (admittedly not with a full-Stokes model), marine ice sheets are often much more challenging for the linear solvers due to the mathematical nature of the floating ice shelves.

My larger objection is that I strongly disagree with the choice of a linear (constant-viscosity) rheology for these experiments. In our experience (again admittedly not with a full-Stokes model), one of the hardest things for many solvers to handle is the large range of viscosities produced by the normal nonlinear rheology. We've often had the case where solvers which perform perfectly well with constant viscosities perform poorly (or fail to converge) when the nonlinear rheology is turned on. I suspect you're getting an incomplete and possibly misleading view of solver performance for "real" ice sheet problems in this case. Is there a compelling reason not to use a "standard" nonlinear rheology for these tests?

Minor points –

1. line 95 – please cite some examples of the full-Stokes solver work that you're refer-ring to

2. line 102 – "well know" -> "well known"

3. line 108 – FS isn't a requirement for active GL dynamics, e.g. MISMIP(1,3d,++). In fact, the authors of this work routinely use SSA for GL problems...

4. line 123 – "suit" -> "suite"

5. line 145 – "period" -> "periodic"

6. line 171 – please elaborate on or clarify what you mean by "methods that naturally

fit the ISSM framework"

7. line 172 – using only the default settings for the PETSc components is likely too-limiting of a choice. I understand why you'd do that (putting yourself in the shoes of a model-user who doesn't want to fiddle with solver parameters or explore all of the options available). However, we've found that there are cases where minor changes in options result in major improvements in solver performance and robustness. I'd suggest that since the goal of this work is to be a reference for ISSM (and other ISM) users, you should make some effort (maybe by asking the PETSc developers or another linear solver expert for some advice) to make the solvers perform as well as possible. I think this work will have a much larger impact in that case. The other point, of course, is that "default" options can change. I'd suggest presenting two sets of results – one with the "default" settings, and one after some attempt has been made to tune the solver parameters. (it is, of course, possible that the default parameters *do* produce the best performance). Of course, then, you would also need to document the particular solver options you used.

8. Conclusion – To give a bit of extra weight to your conclusions, it might be useful to embed your conclusion in the larger context of what many have found to be the case in other scientific computation fields – one suggestion would be to add a statement along the lines of "the conclusion that scalable iterative methods are better suited than direct methods for solving large linear systems echoes the experience of many other researchers across a wide range of scientific disciplines".

9. line 301 – The acknowledgments end with a stray "(" after Jed's name. Perhaps they got cut off?

10. Figure 2 – I am impressed with Figures 2-4 – they do a good job of conveying a lot of information clearly. I'd suggest replacing "horizontal labels" and "vertical labels" with "horizontal rows" and "vertical columns" for clarity in the caption.

11. Figure 6 – It would be helpful to include an "ideal scaling" line in this plot for

comparison. You mention the slopes in the text, but including it on the graph itself can make things easier for the reader.

12. Figure 6 – More numbers than a single "10" on the horizontal axis would also be useful.

13. Figures 5 and 6 – If I read these plots correctly (not completely assured due to the lack of x-axis labeling in Figure 6), it appears that the number of elements per processor used for weak scaling in Figure 5 ($\sim$250) corresponds to the far-right data points (most processors/fewest elements per processor) in the strong-scaling plot in figure 6. In both of the examples, this is where it appears that you start to see a degradation in your solver scaling, which might imply that you're being a bit too aggressive when you generated figure 5 since you seem to have stepped out of your ideal scaling regime. In other words, it might be the case that if you took a look at weak scaling with more elements/processor (500, perhaps), your MUMPS weak scaling might look better.

14. Figure 6 – it would be nice to have one more data point for your strong scaling plots, since it appears that your scaling is just beginning to tail off for MUMPS at the largest number of processors. I also realize that it may be a point too far...

---

## Author Comment (AC1) · 6 Sep 2016

Dear Dr. Astrid Kerkweg,

Thank you for taking the time to check our manuscript that is under review in Geoscientific Model Development Discussions (gmd-2016-111). In accordance to the guidelines you highlighted in your note (SC1), we updated the title in the revised manuscript to include the version number of ISSM used in this study.

Best regards,

Feras Habbal

---

## Author Comment (AC2) · 6 Sep 2016

**Response to RC1:**

In this work, the authors compare the performances of several different direct and iterative solvers, provided by MUMPS and PETSc libraries respectively, for solving the transient ice flow model using ISSM. Specifically, the authors target a well known transient benchmark problem (ISMIP-HOM, test F), in the case of a frozen bed or sliding bed. The flow model is constituted by the ice velocity part (Blatter-Pattyn model with constant viscosity), a part for reconstructing vertical velocities, and the mass transport part. The authors highlight some of the solvers that perform better on different mesh resolutions, for both frozen or sliding bed.
The detailed comparison of the solvers available in ISSM is certainly useful for the several ISSM users. However, I have a few reservations about the impact that this work can have on a broader community.

- The benchmark problem addressed in this work has several simplifications that makes it not very representative of real problems, most notably: 1. Geometry is very simple (in constrast with complex margins or bed roughness encountered in real ice sheets). 2. Viscosity is constant, making the model linear. In real problems viscosity strongly depends on velocity and temperature, which makes the problem much harder to be solved numerically. 3. A relatively high basal friction coefficient is considered, which is not representative of what can be found in ice streams and ice shelves.

Thank you for your comments and review. We chose to test a suite of solvers using a commonly used transient ice flow benchmark test (ISMIP-HOM experiment F), which makes the simplifications that you list, so that other researchers could reproduce the results as well as conduct their own tests using different, potentially customized, solvers or other ice sheet codes with a common and well-known model setup. In addition to updating the manuscript to highlight the limitations of our initial benchmark tests relative to real-world problems, we are including results from applying solvers to another benchmark test (ISMIP-HOM experiment A) in order to explore the impact of using a more realistic nonlinear viscosity model for solving the stress balance equations.

- The authors consider only off-the-shelf solvers that "naturally fit the ISSM framework", whereas several efforts (not mentioned by the authors) have been done in recent years in order to build efficient solvers/preconditioners tailored on the ice sheet problems. Some of these solvers have been demonstrated on large-scale simulations of Greenland or Antarctic ice sheets. See, for example, T. Isaac et al., SIAM J. Sci. Comput., 37(6), B804–B833; Tezaur et al., Procedia Computer Science, 51:2026-2035, ICCS, 2015, S. Cornford at al. J. Comput. Phys, 232(1):529-549, 2013; plus the one by Brown et al. already cited, but not discussed, by the authors. I recommend that the authors make it clear in the abstract that they are only considering the off-the-shelf solvers readily available in ISSM. I also recommend to consider more realistic problems and to mention relevant work in the literature.

We updated the text to include the recommended citations relevant to this work and specified that the focus of this work was to test readily available solvers in the abstract. As you mentioned, our results using PETSc solvers within ISSM are relevant to ISSM users. However, since many numerical models use PETSc, including the Parallel Ice Sheet Model (PISM) and the Community Ice Sheet Model (CISM), which has the ability to leverage PETSc solvers through the Trilinos package, we anticipate that our results should extend to other ice sheet models and benefit modelers beyond the ISSM community.

Minor comments:
- At line 144 the authors mention that they apply single-point constraints on velocity and thickness equations. I'd like the author to expand on this, mentioning how/with what values they constrain in a single point the velocity and thickness. Typically, single-point constraining is used in presence of a singular problems (which should not happen here), and it is known to artificially modify the spectrum of the matrix, which in turn can deteriorate the convergence of iterative solvers.

The text referring to using single point constraints was misstated and is corrected in the updated manuscript to note that we impose Dirichlet boundary conditions. Removing these entries from the matrix does not adversely impact the condition number of the stiffness matrix.

**- Line 150, how the vertical velocity is reconstructed? With an L2 projection?**

We solve the incompressibility equation to recover the vertical velocity, which is constant per element. Subsequently, we use an L2 projection to evaluate the nodal velocity. We updated the manuscript to clarify this point.

**- The time reported in the tables is solver time, or total time (including assembly, linear solvers and I/O)? The weak scaling results for the iterative solvers are not very good and it would be useful to understand what is causing this.**

We updated the text in the manuscript to note that the timing results include solving the system of equations, assembling the stiffness matrix, load vector, and updating the input from the solution.

---

## Author Comment (AC3) · 6 Sep 2016

**Response to RC2:**

This work aims to provide insight into different solver choices for a particular full-Stokes ice sheet model (ISSM) by testing a range of iterative solver choices from the widely available PETSc solver library on a specific test problem (ISMIP-HOM, experiment F) and contrasting with their native/default direct-solve approach (which uses MUMPS). They conclude that switching to the PETSc iterative solvers generally improves time to-solution and scaling as the problem size (number of elements in the finite-element mesh) increases, and are able to provide some suggestions as to which solvers appear to be better suited to their needs. In my opinion, this is a useful contribution to the literature, and I found it to be well-written and well-organized. I do have a few suggestions which I think would greatly increase the usefulness of this work.

My primary criticisms, if you can call them that, are regarding the choice of benchmark problem. While I think that the choice of ISMIP-HOM problem F is a reasonable choice for representing a fully- or mostly-grounded ice sheet (like the Greenland Ice Sheet), I wonder how extendable the results and conclusions are to systems with fast-flowing ice streams and large dynamic ice shelves as are found in Antarctica, represented, for example, by the MISMIP family of benchmarks. In our experience (admittedly not with a full-Stokes model), marine ice sheets are often much more challenging for the linear solvers due to the mathematical nature of the floating ice shelves.

My larger objection is that I strongly disagree with the choice of a linear (constant viscosity) rheology for these experiments. In our experience (again admittedly not with a full-Stokes model), one of the hardest things for many solvers to handle is the large range of viscosities produced by the normal nonlinear rheology. We've often had the case where solvers which perform perfectly well with constant viscosities perform poorly (or fail to converge) when the nonlinear rheology is turned on. I suspect you're getting an incomplete and possibly misleading view of solver performance for "real" ice sheet problems in this case. Is there a compelling reason not to use a "standard" nonlinear rheology for these tests?

Thank you for your review and comments. We used the ISMIP-HOM experiment F test since it involved a transient simulation and is a commonly used benchmark test. The intent of our study was to promote the use of iterative methods over linear solvers using a simplified model, which could then be refined in future work using a real-world simulation. As you mention, the specification of linear viscosity in this benchmark test is a limiting feature in relation to real-world problems. To address the impact of nonlinear rheology on solving the stress balance equations, we are including results from applying the same solvers on another benchmark test (ISMIP-HOM experiment A) that uses a nonlinear viscosity model for ice. We updated the manuscript to highlight the limitations of the transient benchmark test (experiment F) and will include the results from this new study.

**Minor points –**
**1. line 95 – please cite some examples of the full-Stokes solver work that you're referring to**

We added additional references to the manuscript.

**2. line 102 – "well know" -> "well known"**

Fixed typo.

**3. line 108 – FS isn't a requirement for active GL dynamics, e.g. MISMIP(1,3d,++). In fact, the authors of this work routinely use SSA for GL problems...**

We updated the text to avoid implying that full Stokes is required for modeling grounding line dynamics.

**4. line 123 – "suit" -> "suite"**

Fixed typo.

**5. line 145 – "period" -> "periodic"**

Fixed typo.

**6. line 171 – please elaborate on or clarify what you mean by "methods that naturally fit the ISSM framework"**

We updated the manuscript to be clearer on the point that our intention was to use solvers that did not require customization or specialization of the solver routine within ISSM so that the conclusions based on our results could be used by other models as well.

**7. line 172 – using only the default settings for the PETSc components is likely too limiting of a choice. I understand why you'd do that (putting yourself in the shoes of a model-user who doesn't want to fiddle with solver parameters or explore all of the options available). However, we've found that there are cases where minor changes in options result in major improvements in solver performance and robustness. I'd suggest that since the goal of this work is to be a reference for ISSM (and other ISM) users, you should make some effort (maybe by asking the PETSc developers or another linear solver expert for some advice) to make the solvers perform as well as possible. I think this work will have a much larger impact in that case. The other point, of course, is that "default" options can change. I'd suggest presenting two sets of results – one with the "default" settings, and one after some attempt has been made to tune the solver parameters. (it is, of course, possible that the default parameters \*do\* produce the best performance). Of course, then, you would also need to document the particular solver options you used.**

As you summarized, our intention was to highlight strong performance gains that can be attained using iterative solvers in PETSc with limited intervention on the part of modelers (i.e. using default values). In this context, we avoided the large number of options that can be tuned for each combination of iterative scheme with a particular preconditioner and treated each solver with the same level of attention. Also, in light of the simplifications underlying the benchmark test there is no guarantee that speed-ups based on customization of the components would straight forwardly correlate to real-world models. Future work, aimed at refining the results presented in this work, will be based on more realistic models and address the impact of customizing individual components, as you suggested. We updated the manuscript to note that significant performance gains are attainable by customizing the options of the PETSc components for a preferred solver.

**8. Conclusion – To give a bit of extra weight to your conclusions, it might be useful to embed your conclusion in the larger context of what many have found to be the case in other scientific computation fields – one suggestion would be to add a statement along the lines of "the conclusion that scalable iterative methods are better suited than direct methods for solving large linear systems echoes the experience of many other researchers across a wide range of scientific disciplines".**

Indeed this was the main conclusion of this study. We adopted your suggestion and updated the manuscript to emphasize this conclusion.

**9. line 301 – The acknowledgments end with a stray "(" after Jed's name. Perhaps they got cut off?**

Fixed typo.

**10. Figure 2 – I am impressed with Figures 2-4 – they do a good job of conveying a lot of information**

**clearly. I'd suggest replacing "horizontal labels" and "vertical labels" with "horizontal rows" and "vertical columns" for clarity in the caption.**

Thank you for your comments. We updated the figure caption for clarity as you suggested.

**11. Figure 6 – It would be helpful to include an "ideal scaling" line in this plot for comparison. You mention the slopes in the text, but including it on the graph itself can make things easier for the reader.**

As suggested, we updated the figures and captions to denote ideal scaling.

**12. Figure 6 – More numbers than a single "10" on the horizontal axis would also be useful.**

As suggested, we updated the axis bounds.

**13. Figures 5 and 6 – If I read these plots correctly (not completely assured due to the lack of x-axis labeling in Figure 6), it appears that the number of elements per processor used for weak scaling in Figure 5 (~250) corresponds to the far-right data points (most processors/fewest elements per processor) in the strong-scaling plot in figure 6. In both of the examples, this is where it appears that you start to see a degradation in your solver scaling, which might imply that you're being a bit too aggressive when you generated figure 5 since you seem to have stepped out of your ideal scaling regime. In other words, it might be the case that if you took a look at weak scaling with more elements/processor (500, perhaps), your MUMPS weak scaling might look better.**

While using 250 elements per processor provided the fastest results for iterative methods applied to all but the largest model, your assessment that the scaling deteriorates at larger model sizes, especially for the linear solver is correct. Also, figure 5 had an error in color scale, which misrepresented the results. We fixed this error in this figure and used 500 elements/CPU for presenting weak scaling, as you suggested.

**14. Figure 6 – it would be nice to have one more data point for your strong scaling plots, since it appears that your scaling is just beginning to tail off for MUMPS at the largest number of processors. I also realize that it may be a point too far...**

The number of points used for scaling was chosen to be consistent with the tests that were performed for all solvers and plotted in Figures 2-4.

---

## Referee Report (RR1)

The work "Optimal numerical solvers for transient simulations of ice flow using the Ice Sheet System Model (ISSM)" by Habbal et al. has been improved during the revision process. In particular the authors explained better the scope of the work, improved the description of the methods and results, and added a benchmark (ISMIP HOM test A) to investigate the impact of nonlinear viscosity. However, they did not address other comments of the referees. Most importantly they did not studied the performance of the different solvers/preconditioners in the case of fast sliding regimes (occurring e.g. on ice shelves), which are known to be particularly challenging for standard solver/preconditioners. For this reason I would encourage the authors to try solving test A (with same aspect ratio) replacing the no-slip condition at the bed with a sliding condition with very small friction coefficient.

In the numerical results, total computational times are reported. However, given that the focus of the paper is on linear solvers and preconditioners, it would be useful if the authors reported, at least for a selection of  solvers, also the computational time for the liner solvers/preconditioners.

I would like to bring to the attention of the authors a paper published very recently that might be relevant to this work: "A Matrix Dependent/Algebraic Multigrid Approach for Extruded Meshes with Applications to Ice Sheet Modeling" by Tuminaro et. al, *SIAM J. Sci. Comput.*

---

## Author Response (AR2)

The work "Optimal numerical solvers for transient simulations of ice flow using the Ice Sheet System Model (ISSM)" by Habbal et al. has been improved during the revision process. In particular the authors explained better the scope of the work, improved the description of the methods and results, and added a benchmark (ISMIP HOM test A) to investigate the impact of nonlinear viscosity. However, they did not address other comments of the referees. Most importantly they did not studied the performance of the different solvers/preconditioners in the case of fast sliding regimes (occurring e.g. on ice shelves), which are known to be particularly challenging for standard solver/preconditioners. For this reason I would encourage the authors to try solving test A (with same aspect ratio) replacing the no-slip condition at the bed with a sliding condition with very small friction coefficient.

Thank you for your review of this paper. We added results for Experiment A to the manuscript, which involved a significant number of simulations and computational resources, to highlight the impact of using nonlinear viscosity on the solver performance. In this regard, Experiment A was a significantly different test from the ISMIP-HOM Experiment F benchmark test. However, due to the limited allocation of computational resources, we did not consider alternate basal boundary conditions of Experiment A, since we included results from Experiment F across both the sliding and no-slip boundary conditions and highlighted the methods which performed the fastest across these distinct boundary conditions over the broadest mesh size (i.e. number of elements).

In the numerical results, total computational times are reported. However, given that the focus of the paper is on linear solvers and preconditioners, it would be useful if the authors reported, at least for a selection of solvers, also the computational time for the liner solvers/preconditioners.

The computational times, provided by the ISSM profiling tools, are used to compare the relative differences between the tested methods. Since these steps are independent of the given method, these steps do not bias the timing results.

I would like to bring to the attention of the authors a paper published very recently that might be relevant to this work: "A Matrix Dependent/Algebraic Multigrid Approach for Extruded Meshes with Applications to Ice Sheet Modeling" by Tuminaro et. al, SIAM J. Sci. Comput.

Thank you for providing the reference to this recent work, which we have added to the manuscript.

[revised manuscript text omitted]